# PERSPECTIVE

## Shaping the waves: Mitochondrial regulation of calcium oscillations in smooth muscle

Harold A. Coleman [ID]
and Helena C. Parkington [ID]

*Department of Physiology, Biomedicine Discovery Institute, Monash University, Clayton, VIC, Australia*

Email: harry.coleman@monash.edu

Handling Editors: Bjorn Knollmann & Nikki Jernigan

The peer review history is available in the Supporting Information section of this article (https://doi.org/10.1113/JP288974#support-information-section).

That cytoplasmic free calcium is a critical player in many cellular processes has been known for decades. The major enduring question is how this one ion is able to activate diverse processes within the same cell with some degree of specificity.

Microdomains could provide a good explanation, of which membrane invaginations, caveolae, in association with the endoplasmic reticulum (ER, a major calcium store), provide a conceptually understandable basis, particularly since a gap of only ∼20 nm separates the plasmalemmal caveolae and the ER. Compelling imaging and functional studies indicate that calcium movement pathways cluster together in these microdomains resulting in expeditious signalling (Akin et al., 2023). Also, physical tethering between caveolar and ER proteins stabilized this niche. Another microdomain occurs between the ER and mitochondria, which also includes physical tethering, enabling complex interactions (Cartes-Saavedra et al., 2025).

An intriguing question is the different roles played by the various sources of calcium. In smooth muscle, waves of intracellular calcium can propagate between cells. The function(s) of these waves and their relationship to the electrical and contractile activity of smooth muscle are not clear. Action potentials occur in near synchrony throughout the cell resulting in a global increase in calcium as a result of influx through voltage-gated calcium channels (VGCCs). In many smooth muscles, it is this calcium that is responsible for contractions (Gravina et al., 2010) while in some others, VGCC-mediated calcium influx evokes the release of calcium from intracellular stores which then activates contraction or provides calcium for refilling the store (Akin et al., 2023).

The ER and mitochondria come into close contact. In this microdomain, the calcium-conducting $IP_3$ receptors in the ER membrane are tethered to the abundant mitochondrial VDAC-1 ion channels via the GrP75 linker protein. VDAC conducts ions, including calcium. Mitochondrial calcium-activated dehydrogenase increases NADH and ATP production (Cartes-Saavedra et al., 2025). This system plays a major role in heart function. In some smooth muscles, mitochondria may exert a modulatory role in both pace making and contractile activity (Gravina et al., 2010).

To better understand mitochondrial function in smooth muscle, an elegant study by Yamamura et al (2018) used isolated cells to use sophisticated imaging techniques coupled with patch clamp electrophysiology. The results provide valuable insights into the voltage dependence and time course of stimulus-evoked changes in cytoplasmic and mitochondrial calcium levels and provide evidence that mitochondria are functionally coupled to the ER. In smooth muscle tissue, the cells are normally coupled to each other and, in some cases, also to other cell types such as endothelial and interstitial cells, via gap junctions. This coupling precludes the use of some techniques. Nevertheless, there remains an important question: what happens in intact tissues, where signals can propagate between cells and cell types?

Significantly, in this issue of *The Journal of Physiology*, Zhang et al (2025) studied intact tissues to investigate interactions between the ER and mitochondria in vascular smooth muscle. The main technique involved imaging of cytoplasmic free calcium and changes in mitochondrial membrane potential, combined with various pharmacological agents. Tissues were depolarized with high-$K^+$ physiological saline and this evoked oscillations in intracellular calcium. To limit tissue movement for imaging, contractions were prevented by wortmannin which inhibits myosin light chain kinase, a technique used in many studies of smooth muscle. However, nanomolar concentrations of wortmannin inhibit phosphoinositide 3-kinases and this has implications for the production of $IP_3$, which opens the ER calcium store, and $PIP_2$, which can strongly influence the activity of many ion channels that can change membrane potential and hence VGCC activity, as discussed by Akin et al (2023). Smooth muscle depolarization evokes calcium oscillations as a result of the activity of VGCCs, $K_V$, $K_{Ca}$ and ANO1 channels. In the present study, the 30 mM high-$K^+$ would shift $E_K$ to ∼−43 mV, just below the expected threshold for action potential activation and, although oscillations in membrane potential were limited, they cannot be dismissed. The recording of membrane potential together with contractile activity would be a difficult but important indication of whether membrane potential oscillations were occurring.

A key observation by Zhang et al (2025) is that the calcium response to high $K^+$ could be separated into a slow component, due to calcium influx through VGCCs, and a fast oscillating component attributed to calcium release from the ER via $IP_3$ receptors. It was concluded that high $K^+$ activated $IP_3$ receptors via the VGCC-sourced calcium, an interesting interpretation that requires further investigation. To assess the contribution of mitochondria to the calcium oscillations, the uncoupler carbonyl cyanide 3-chlorophenylhydrazone (CCCP) was applied to prevent calcium uptake by mitochondria. Intriguingly, CCCP inhibited the calcium oscillations. The antioxidant MitoTEMPO showed that reactive oxygen species production by mitochondria was involved in ER/mitochondria interactions.

A strength of this study is that it investigated intact tissues rather than isolated cells. The results form a cohesive story in which VGCC-mediated calcium influx activates two different calcium signals. Accordingly, some calcium becomes bulk calcium to activate the contractile machinery while other calcium activates the release of pulsatile calcium from the ER/mitochondria, suggesting an important role for microdomain(s). If the mitochondria are unable to take up calcium, then the fast oscillations in calcium do not occur. Significantly, this study shows that

the mitochondria are critical for the release of calcium from the ER via IP$_3$ receptors, thus shaping this calcium signal. However, as with most studies, there is scope for additional investigations, particularly if more selective ways of inhibiting some of the processes could be applied.

In throwing light on complex ER/mitochondrial interactions in intact tissues, this study raises intriguing questions. First, the mechanism(s) by which the mitochondria interact with the ER to enable fast oscillations in cytoplasmic calcium are far from clear. Second, the relationships between the ER, mitochondria, and caveolar microdomains require clarification. Broader questions include the function(s) of the fast calcium oscillations. It is hoped that the results of the study by Zhang et al (2025) will stimulate further experiments to shed additional light on such issues.

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

## Additional information

### Competing interests

No competing interests declared.

## Author contributions

H.A.C. and H.C.P. were responsible for the conception or design of the work, as well as drafting the work or revising it critically for important intellectual content. Both authors approved the final version of the manuscript submitted for publication and agree to be accountable for all aspects of the work in ensuring that questions related to the accuracy or integrity of any part of the work are appropriately investigated and resolved. All persons designated as authors qualify for authorship, and all those who qualify for authorship are listed.

## Funding

HCP's research was funded by the National Health & Medical Research Council of Australia, grant Gnt2000774, and the Australian Research Council, grant DP230101068.

## Keywords

calcium store, calcium wave, endoplasmic reticulum, intracellular calcium, IP3, mitochondria, smooth muscle, voltage-dependent calcium channel

## Supporting information

Additional supporting information can be found online in the Supporting Information section at the end of the HTML view of the article. Supporting information files available:

**Peer Review History**

