## [Peer Review History · The Journal of Physiology]

Shaping the waves – mitochondrial regulation of calcium oscillations in smooth muscle

Harold A Coleman and Helena C. Parkington

DOI: 10.1113/JP288974

Corresponding author(s): Harold Coleman (harry.coleman@monash.edu)

Review Timeline:

Submission Date:	11-Apr-2025
Editorial Decision:	16-Apr-2025
Revision Received:	22-Apr-2025
Accepted:	30-Apr-2025

Senior Editor: Bjorn Knollmann

Reviewing Editor: Nikki Jernigan

Transaction Report:

Dear Dr Coleman,

Re: JP-P-2025-288974 "**Shaping the waves - mitochondrial regulation of calcium oscillations in smooth muscle**" by Harold A Coleman and Helena C. Parkington

Thank you for submitting your manuscript to The Journal of Physiology. It has been assessed by a Reviewing Editor and by an expert referee and we are pleased to tell you that it is acceptable for publication following satisfactory revision.

The review comments are copied at the end of this email.

Please address all the points raised and incorporate all requested revisions or explain in your Response to Referees why a change has not been made. We hope you will find the comments helpful and that you will be able to return your revised manuscript within 2 weeks. If you require longer than this, please contact journal staff: jp@physoc.org.

REVISION CHECKLIST:

We look forward to receiving your revised submission.

Yours sincerely,

Bjorn Knollmann
Senior Editor
The Journal of Physiology

EDITOR COMMENTS

Reviewing Editor:

Just one minor revision on the description/terminology of voltage-gated Ca channel activation.

Senior Editor:

I concur with the reviewing editor. Excellent piece, just needs a minor correction as outlined by reviewer 1

REFEREE COMMENTS

Referee #1:

The perspective by Coleman & Parkington begins by outlining several unresolved questions regarding the regulation of cell function by Ca^{2+} and mitochondria. The perspective then provides a summary of the Zhang et al. (2025) study, highlighting the key findings and suggesting directions for future research. The perspective is well-written and clear. One minor point worth noting is the description of a 'threshold' for voltage-dependent Ca^{2+} channel activity in the perspective. While this is a common description, there is no threshold for opening. The activation of L-type voltage-dependent Ca^{2+} channels (CaV1.2) follows an exponential relationship with membrane potential and shows an e-fold increase in open probability with approximately every 7-8 mV of depolarization. The overall voltage-dependence follows a sigmoidal (Boltzmann) curve, rather than a sharp threshold or step function.

END OF COMMENTS

Response to Referees

.....

EDITOR COMMENTS

Reviewing Editor:

Just one minor revision on the description/terminology of voltage-gated Ca channel activation.

Senior Editor:

I concur with the reviewing editor. Excellent piece, just needs a minor correction as outlined by reviewer 1

Comment: We appreciate the positive response to our manuscript and the opportunity to correct the issue quite rightly raised by Referee #1.

.....

REFEREE COMMENTS

Referee #1:

The perspective by Coleman & Parkington begins by outlining several unresolved questions regarding the regulation of cell function by Ca^{2+} and mitochondria. The perspective then provides a summary of the Zhang et al. (2025) study, highlighting the key findings and suggesting directions for future research. The perspective is well-written and clear. One minor point worth noting is the description of a 'threshold' for voltage-dependent Ca^{2+} channel activity in the perspective. While this is a common description, there is no threshold for opening. The activation of L-type voltage-dependent Ca^{2+} channels (CaV1.2) follows an exponential relationship with membrane potential and shows an e-fold increase in open probability with approximately every 7-8 mV of depolarization. The overall voltage-dependence follows a sigmoidal (Boltzmann) curve, rather than a sharp threshold or step function.

Reply: We agree with the Referee's comments that the term threshold does not apply to the activation of voltage-gated calcium channels. Rather, the term threshold is more appropriate for the activation of action potentials. We have therefore replaced "VGCC activity" with "action potential activation" (highlighted in lines 60 – 61).

END OF COMMENTS

Dear Dr Coleman,

Re: JP-P-2025-288974R1 **"Shaping the waves - mitochondrial regulation of calcium oscillations in smooth muscle"**
by Harold A Coleman and Helena C. Parkington

We are pleased to tell you that your paper has been accepted for publication in The Journal of Physiology.

Yours sincerely,

Bjorn Knollmann
Senior Editor
The Journal of Physiology

If you would like to receive our 'Research Roundup', a monthly newsletter highlighting the cutting-edge research published in The Physiological Society's family of journals (The Journal of Physiology, Experimental Physiology, Physiological Reports, The Journal of Nutritional Physiology, and The Journal of Precision Medicine: Health and Disease), please click this link, fill in your name and email address and select 'Research Roundup':

<https://www.physoc.org/journals-and-media/membernews>

- You can help your research get the attention it deserves! Check out Wiley's free Promotion Guide for best-practice recommendations for promoting your work at: www.wileyauthors.com/eeo/guide. You can learn more about Wiley Editing Services which offers professional video, design, and writing services to create shareable video abstracts, infographics, conference posters, lay summaries, and research news stories for your research at: www.wileyauthors.com/eeo/promotion.

The Corresponding Author will receive an email from Wiley with details on how to register or log-in to Wiley Authors Services where you will be able to place an order

EDITOR COMMENTS

Reviewing Editor:

The authors have addressed all comments

Senior Editor:

Excellent contribution to the Journal!

REFeree COMMENTS

Referee #1:

The authors have fully addressed my very minor comments